# Pathological Features and Genomic Characterization of an *Actinobacillus equuli* subsp. *equuli* Bearing Unique Virulence-Associated Genes from an Adult Horse with Pleuropneumonia

**DOI:** 10.3390/pathogens12020224

**Published:** 2023-01-31

**Authors:** Maedeh Kamali, Mariano Carossino, Fabio Del Piero, Laura Peak, Maria S. Mitchell, Jackie Willette, Rose Baker, Fuyong Li, Ákos Kenéz, Udeni B. R. Balasuriya, Yun Young Go

**Affiliations:** 1Department of Infectious Diseases and Public Health, Jockey Club College of Veterinary Medicine, City University of Hong Kong, Hong Kong SAR, China; 2Louisiana Animal Disease Diagnostic Laboratory, School of Veterinary Medicine, Louisiana State University, Baton Rouge, LA 70803, USA; 3Department of Pathobiological Sciences, School of Veterinary Medicine, Louisiana State University, Baton Rouge, LA 70803, USA; 4Veterinary Teaching Hospital and Department of Veterinary Clinical Sciences, School of Veterinary Medicine, Louisiana State University, Baton Rouge, LA 70803, USA

**Keywords:** *Actinobacillus equuli* subsp. *equuli*, equine actinobacillosis, whole genome sequencing

## Abstract

*Actinobacillus equuli* subsp. *equuli* is the etiological agent of sleepy foal disease, an acute form of fatal septicemia in newborn foals. *A. equuli* is commonly found in the mucous membranes of healthy horses’ respiratory and alimentary tracts and rarely causes disease in adult horses. In this study, we report a case of a 22-year-old American Paint gelding presenting clinical signs associated with an atypical pattern of pleuropneumonia subjected to necropsy. The gross and histopathological examinations revealed a unilateral fibrinosuppurative and hemorrhagic pleuropneumonia with an infrequent parenchymal distribution and heavy isolation of *A. equuli*. The whole genome sequence analysis indicated that the isolate shared 95.9% homology with the only other complete genome of *A. equuli* subsp. *equuli* available in GenBank. Seven virulence-associated genes specific to the isolate were identified and categorized as iron acquisition proteins, lipopolysaccharides (LPS), and capsule polysaccharides. Moreover, four genes (*glf*, *wbaP*, *glycosyltransferase family 2 protein*, and *apxIB*) shared higher amino acid similarity with the invasive *Actinobacillus* spp. than the reference *A. equuli* subsp. *equuli* genome. Availability of the whole genome sequence will allow a better characterization of virulence determinants of *A. equuli* subsp. *equuli*, which remain largely elusive.

## 1. Introduction

*Actinobacillus* species (spp.) are gram-negative, anaerobic, and pleomorphic bacteria in the family *Pasteurellaceae* [1]. *Actinobacillus* spp. reside on the mucosal membranes of healthy animals’ respiratory and genitourinary tracts [2]. Clinical manifestations caused by these organisms vary, including respiratory infections, septicemia, metritis, mastitis, arthritis, endocarditis, meningitis, or stillbirth [3,4,5,6,7,8,9]. In horses, *A. equuli* subspecies (subsp.) *equuli* or *A. equuli* subsp. *haemolyticus* are commonly found in the normal oral flora of healthy horses and the alimentary and genital tracts, whereas *A. ligneieresii* and *A. pleuropneumoniae* are less frequently found [10,11,12]. Interestingly, *A. equuli* subsp. *equuli* differs from *A. equuli* subsp. *haemolyticus* by manifestation of different disease patterns and epidemiology. *A. equuli* subsp. *equuli* is primarily associated with “sleepy foal disease”, an acute form of fatal septicemia in newborn foals that can develop into a chronic state with kidney, joint, and lung lesions [3,13,14]. Although *A. equuli* rarely causes disease in adult horses, a few acute and chronic cases of peritonitis have been previously reported [4,15,16]. In addition, it has been recently reported that *A. equuli* acts as a primary pathogen in sows and piglets, given favorable conditions [13,14]. *Actinobacillus equuli* subsp. *equuli* is non-hemolytic and CAMP-negative, which are the main features that differentiate from *A. equuli* subsp. *haemolyticus* [13]. *Actinobacillus equuli* subsp. *equuli* does not contain repeats-in-toxins (RTX) activity referred to as the *A. equuli* toxin (*Aqx*) or any other related RTX genes, which confers the hemolytic phenotype associated with pulmonary hemorrhage and endothelial damage in an adult horse [15]. Although several virulence factors associated with pathotypes of certain *Actinobacillus* spp. have been identified, such as autotransporters (ATs), outer membrane proteins (OMPs), capsule polysaccharide (CPS), lipopolysaccharide (LPS), iron acquisition mechanisms, RTX, and urease, the virulence factors associated with *A. equuli* subsp. *equuli* are largely unknown [2,15,16,17]. We hypothesized that a comprehensive genome analysis of *A. equuli* subsp. *equuli* isolated from a clinical case can improve our understanding of the potential invasion determinants and the virulence factors associated with the pathogenicity and disease phenotype.

Here, we describe a clinical case of *A. equuli* subsp. *equuli* associated with an atypical pattern of pleuropneumonia in an adult horse, in which this infection is usually rare or sporadic. The gross and histopathological examinations revealed lesions in the left lung consistent with fibrinosuppurative and hemorrhagic pleuropneumonia associated with heavy isolation of *A. equuli*. To further understand the basis of the disease phenotype, full-length genome sequence of the isolate was obtained and characterized. Availability of the complete genome sequence will allow a better characterization of virulence determinants of *A. equuli* subsp. *equuli*, which remain largely elusive.

## 2. Materials and Methods

### 2.1. Postmortem Examination and Sample Collection

A postmortem examination of a 22-year-old American Paint horse with atypical patterns of pleuropneumonia was performed at the Louisiana Animal Disease Diagnostic Laboratory (LADDL). Specimens from the left lung were aseptically collected and submitted for additional ancillary testing (bacteriological culture). Samples from all organs were fixed in 10% neutral buffered formalin for 24–48 h for histopathologic examination.

### 2.2. Histopathology

Formalin-fixed tissues (lung, liver, adrenal glands, spleen, kidneys, heart, encephalon [cerebrum, cerebellum, brainstem], pituitary gland, mesenteric lymph node, pancreas, and gastrointestinal tract) were embedded in paraffin. Subsequently, formalin-fixed paraffin-embedded (FFPE)-tissue sections (4 μm) were stained with hematoxylin and eosin (H&E) according to a standard laboratory procedure before histological evaluation.

### 2.3. Bacteriology, MALDI-TOF Mass Spectrometry, and Antibiotic Sensitivity

Samples from the lungs were subjected to routine bacteriological testing by inoculating commercial blood agar, chocolate agar, and MacConkey agar plates (Remel, San Diego, CA, USA). Inoculated blood and chocolate agar plates were incubated at 37 °C with 5% CO_2_ for the first 24 h and without CO_2_ for the next 48 h. Inoculated MacConkey agar plates were incubated at 37 °C for up to 72 h. Bacterial colonies were subsequently screened using a MALDI Biotyper^®^ Compass (Bruker, Billerica, MA, USA) [(automated identification system based on matrix-assisted laser desorption/ionization time of flight (MALDI-TOF)] for identification using the direct transfer method. Identification was further clarified with phenotypic presentation combined with Christie–Atkins–Munch-Peterson (CAMP) test results using pure cultures following standard procedures. Antibiotic sensitivities were performed using the broth microdilution method (Sensititre™ EQUIN1F Vet AST Plate) to determine minimum inhibitory concentrations (MICs).

### 2.4. DNA Extraction, Whole Genome Sequencing, and Assembly

Genomic DNA was extracted from pure bacterial colonies using Qiagen’s DNeasy Blood and Tissue Kit (Qiagen, Germantown, MD, USA) on a QIAcube automated workstation (Qiagen). DNA concentrations were determined using the Qubit dsDNA HS Assay Kit on a Qubit flex fluorometer (ThermoFisher Scientific, Waltham, MA, USA). DNA libraries were constructed with the Nextera XT DNA Library Preparation Kit (Illumina^®^, San Diego, CA, USA) using 0.2 ng/µL of genomic DNA. Resultant libraries were sequenced on an Illumina MiSeq desktop sequencer using v2 sequencing chemistry with 2 × 250 bp pair-end reads per the manufacturer’s protocol. The quality control of raw sequences was performed by FastQC v0.11.9, and the de novo assembly was done using SPAdes v3.14.1. The final contigs of the isolated strain (designated strain 4524) were subjected to BLASTn to identify the reference genome, which shows the highest identity with the contigs. The genome of the 4524 strain was reordered using progressive Mauve with the *A. equuli* subsp. *equuli* ATCC 19392 genome as the reference genome [18]. The FASTA file of the 4524 genomes was used for genome annotation and further analysis.

### 2.5. A. equuli Subsp. equuli Genome Annotation and Analysis

The whole genome sequence of the 4524 strain was annotated using Bakta v.1.3.0, Prokka v.1.12, and NCBI Prokaryotic Genome Annotation Pipeline [19,20]. The proteome of the 4524 strain was subjected to BLASTp against the Cluster of Orthologous Groups (COGs) database at E-value < 1^e−5^ to identify the COGs [21]. The CRISPI web interface with parameters set to the default values was used to find CRISPR sequences in the genomes [22]. The CGView Server [23] (http://stothard.afns.ualberta.ca/cgview_server/ (accessed on 12 May 2022) was used to generate the circular genome map of the 4524 strain with *A. equuli* subsp. *equuli* ATCC 19392 as the reference genome. The GenBank files of *A. equuli* subsp. *equuli* ATCC 19392 and 4524 strains were submitted to EDGAR for comparative genomic analyses [24]. The EDGAR private project was developed for generating Venn diagrams and verifying shared genes and core genomes between the two strains. Virulence-associated genes described in previous studies were identified through the genome of the 4524 strain and aligned to the homologous genes in the ATCC 19392 genome.

### 2.6. Phylogenetic Analysis

Thirteen *Actinobacillus* bacteria belonging to various species were selected, and their respective nucleotide sequences were retrieved from the GenBank database. The selected strains and their characteristics are presented in Appendix A. Whole-genome alignments were performed using REALPHY (http://realphy.unibas.ch (accessed on 20 June 2022)) [25]. The phylogenetic tree was constructed using the maximum likelihood method, and the evolutionary distances were computed using the general time-reversible model along with gamma-distributed rates (G = 0.6) and invariant site (I) by MEGA v.7 [26,27,28]. Branch validity was evaluated by the bootstrap test with 1000 replications. The average nucleotide identity (ANI) values of selected *Actinobacillus* spp. strains were calculated using the server EzBioCloud (http://www.ezbiocloud.net/tools/ani (accessed on 1 July 2022)) [29]. According to the algorithm developed by Goris et al. [30], a 95∼96% cut-off value was used for the species boundary [31]. The web-based DSMZ service (http://ggdc.dsmz.de (accessed on 3 July 2022)) [32] with 70% species and sub-species cut-off was used to estimate the in silico genome-to-genome distance values for the 4524 isolate and selected strains.

## 3. Results

### 3.1. Case Presentation

In March 2020, a 22-year-old American Paint gelding presented to the Louisiana State University Veterinary Teaching Hospital due to acute onset of illness with tachypnea, bilateral purulent to hemorrhagic nasal discharge, and cough with bloody sputum. The gelding was kept at a boarding facility and used as a lesson horse. The illness presented suddenly, and he was brought immediately to the Veterinary Teaching Hospital for evaluation and treatment due to the rapid onset and progression of clinical signs. The horse had no previous illness and was not receiving treatment for any other known underlying condition. Due to a possible infectious disease concern, the gelding was immediately admitted to the isolation facility. On presentation, major findings included a marked leukocytosis (12.1 × 10^3^/μL [reference range: 5.0–11.0 × 10^3^/μL) with left shift (5.8% bands; 0.7 × 10^3^/μL [reference range: 0.0–0.1 × 10^3^/μL]), toxic neutrophils, and monocytosis (12%; 1.5 × 10^3^/μL [reference range: 0.0–0.8 × 10^3^/μL]). Ultrasonographically, pleural effusion and pulmonary consolidation of the left lung were noticed. A thoracocentesis revealed neutrophilic exudation with short, intracytoplasmic bacterial rods, indicating marked septic neutrophilic inflammation. The direct smears of pleural fluid contained moderate numbers of nucleated cells and erythrocytes. The nucleated cells comprised approximately 70% of non-degenerate neutrophils, 30% of macrophages, and rarely small lymphocytes. In addition, a few neutrophils containing phagocytosed short bacilliform bacteria were observed. The macrophages were variably vacuolated, with some cells having more rounded borders and increased nuclear-to-cytoplasmic ratios. Due to the overall poor prognosis, humane euthanasia was elected by the owner, and the horse was submitted for a thorough postmortem examination.

### 3.2. Gross and Histopathological Findings

Macroscopically, an extensive area (approximately 50 cm in diameter) of the pulmonary parenchyma at the level of the caudoventral aspect of the left lung was dark red and firm, with the lining pleura overlaid by a thick layer of fibrin (Figure 1). The cranial part of the left lung also had multiple dark red pin-point foci. There was no evidence of exudate within nasal passages, paranasal sinuses, guttural pouches, laryngeal, tracheal, or bronchial lumina. The abdominal cavity contained approximately 0.5–1 L of yellow-tinged fluid. No other significant gross abnormalities were found.

Histologically, the affected parenchyma was characterized by liquefactive necrosis and fibrinosuppurative inflammation with loss of alveolar septa and/or alveoli diffusely filled with necrotic debris abundant of degenerate neutrophils, fibrin exudation, edema, and hemorrhage. Bronchioles were frequently filled with large necrotic cell debris and degenerate neutrophils, and numerous, variably sized blood vessels were occluded by fibrin thrombi. The interlobular septa were distended by edema, fibrin exudation, and hemorrhage. The pleura was significantly expanded by abundant fibrin exudation and degenerate neutrophils and coated by a thick layer of polymerized fibrin and myriad colonies of short, gram-negative bacterial rods (Figure 2). Additional findings included fibrinous capsulitis and tubulointerstitial nephritis, and mild neutrophilic hepatitis. Overall, the gross and histological changes in the left lung were consistent with fibrinosuppurative and hemorrhagic pleuropneumonia associated with heavy isolation of *A. equuli*, in agreement with the clinical history and differential diagnosis considered at the time of euthanasia.

### 3.3. Bacteriology, MALDI-TOF Mass Spectrometry, and Antibiotic Sensitivity

Initially, MALDI-TOF identified the organism as *Actinobacillus suis* with a score of 2.47. However, the MALDI patterns and the 16S rRNA gene sequences of *A. arthritidis*, *A. capsulatus*, *A*, *equuli*, *A. lignieresii*, *A. pleuropneumoniae*, and *A. suis* are highly similar; therefore, they cannot be distinguished from each other. Since *A. suis* did not fit with the case, we used the growth patterns and CAMP test to speciate further. The isolate was non-hemolytic on blood agar with no growth on MacConkey agar. In addition, the CAMP test was negative. Using this additional information, the isolate was identified as *A. equuli.* Finally, the isolate was submitted for WGS for further confirmation of the identification. Broth microdilution minimum inhibitory concentration (MIC) results can be seen in Table 1.

### 3.4. Genome Properties of the A. equuli subsp. equuli 4524 Isolate

Whole-genome sequencing was performed to characterize the genetic properties of the *A. equuli*, 4524 strain, isolated from this clinical case. The assembled genome contained 118 contigs, with N50 of 213,479 bp and approximately 43× sequence coverage. The full-length genome was obtained with a length of 2,464,726 bp, a G + C content of 40.2%, and 2362 predicted genes consisting of 2277 protein-coding genes, and 85 for RNA (61 tRNA, 1 tmRNA, 8 rRNA, and 15 ncRNA genes) (Table 2). Approximately 60% of the predicted genes were assigned to 26 functional COG categories (Table 3).

The phylogenetic analysis using the whole genome sequences of *Actinobacillus* spp. identified the 4524 isolate as *A. equuli* subsp. *equuli* owing to clustering together with *A. equuli* subsp. *equuli* ATCC 19392 strain (Figure 3).

The genome of the isolate displayed the highest average nucleotide identity (ANI) value (95.9%) and the lowest genome-to-genome distance calculation (GGDC) values (0.0418) to the genome of *A. equuli* subsp. *equuli* ATCC 19392 (Table 4). Notably, the highest DNA-DNA hybridization (DDH) value (66.1) was measured between the whole genomes of the 4524 and ATCC 19392 strains, with ≥70% probability indicating that these two strains belong to the same species (Table 3). DDH value ≤ 70% is interpreted as two distinct species, while DDH value ≥ 70% is inferred as two tested organisms belonging to the same species.

Further, the BLASTn comparison of the genome sequences between 4524 and ATCC 19392 strains using the CGView Comparison Tool (CCT) showed they are very similar, suggesting the isolate belongs to *A. equuli* subsp. *equuli* (Figure 4).

### 3.5. Comparative Genome Analysis of A. equuli Subsp. equuli 4524 with ATCC 19392 Strain

The genome of *A. equuli* subsp. *equuli* 4524 was compared with ATCC 19392 strain using EDGAR pipeline. The EDGAR pipeline found 2483 CDS in the genomes of 4524 and ATCC 19392 strains, of which the two strains shared 1969 CDS (79.3%). Venn diagram of the two strains showed that there are 302 genes unique to isolate 4524 while 212 genes were specific to the ATCC 19392 strain (Figure 5).

In addition, 136 putative virulence genes previously identified in other *Actinobacillus* spp. were detected in the genome of both strains, of which 122 were present in both strains. In comparison, seven virulence-associated genes were only detected in the ATCC 19392 genome, and seven genes were only identified in the genome of 4524 isolate (Table 5, Appendix A).

Of 129 virulence-associated genes identified in the genome of 4524 strain, 72 genes encoded RTX toxins, lipopolysaccharides (LPS), capsular polysaccharide (CPS), outer membrane protein (OMP), salicylic acid, and biofilm (*pgaA*, *pgaB*, *pgaC*). In addition, several putative iron acquisition systems, including those for hemoglobin (*hgbA*), transferrin (TonB and TonB-*exbBD*), and siderophore (*fhuBCD*), were detected in the genome. In addition, different types of adhesins, including filamentous hemagglutinin (*fhaC*), tight adherence protein (*tadA-F*), fimbria-like protein (*flpB*, *flpC*), and autotransporters that may be involved in adhesion of the bacteria to the target cells were also found (Appendix A). Among the strain-specific genes of *A. equuli* subsp. *equuli* 4524 strain, three were categorized as iron acquisition proteins (*TonB-dependent receptor plug domain protein*, *FhuD*, and *FhuB*), two LPS (*glycosyltransferase family 2 protein* and *rfaF*), one CPS (*putative glycosyltransferase YkoT*), and one miscellaneous (*pgaC*) (Appendix A). The shared genes between *A. equuli* subsp. *equuli* 4524 and ATCC 19392 strains showed > 90% nucleotide similarity, except for three genes categorized as LPS (*glf*, *wbaP*, and *glycosyltransferase family 2 protein)* and the *apxIB* gene encoding toxin exporter. The three LPS genes, including two *glf* genes and *WbaP*, encoded protein sequences of 94% similarity with homologous proteins present in *A. pleuropneumoniae* and *A. suis*, respectively (Appendix A). Another LPS gene encoding glycosyltransferase family 2 protein was 99.3% identical to the homologous protein in the genome of *A. ureae* (Appendix A). The LPS gene transfers activated sugar residue from a donor substrate to a proper acceptor [33]. Interestingly, the *apxIB* gene synthesizing RTX toxin exporter in the 4524 strain showed only 65% similarity at both nucleotide and amino acid levels compared to that of ATCC 19392 strain. In contrast, it was 99.7% and 99.9% similar at the nucleotide and amino acid levels to the *aqxB protein* gene of *A. equuli* subsp. *haemolyticus* strain. (Appendix A).

## 4. Discussion

*Actinobacillus equuli* subsp. *equuli* has traditionally been associated with neonatal septicemia. In contrast, infections in adult horses are less common and often associated with concurrent bacterial or viral infections as well as other predisposing factors such as drug-associated damage or stressful events that compromise the mucosal integrity [7,12,15,16]. In this study, we described a case of a 22-year-old American Paint gelding that suffered respiratory distress with tachypnea, bilateral purulent to hemorrhagic nasal discharge, and cough with bloody sputum. In agreement with the clinical history and differential diagnosis considered at the time of euthanasia, the gross and histopathological changes observed in the left lung were consistent with fibrinosuppurative and hemorrhagic pleuropneumonia associated with heavy isolation of *A. equuli*. Further whole genome sequencing and phylogenetic analyses of the isolate identified *A. equuli* subsp. *equuli* as the causative agent of fibrinosuppurative and hemorrhagic pleuropneumonia. *Actinobacillus equuli* is classified into two subspecies: *A. equuli* subsp. *equuli* and *A. equuli* subsp. *haemolyticus*. *Actinobacillus equuli* subsp. *equuli* usually is innocuous in the oral cavity and alimentary tract of adult horses, whereas *A. equuli* subsp. *haemolyticus* resides preferentially in the respiratory tract and is frequently isolated from the normal oral cavity and tracheal wash fluids [3,13,14]. In contrast to this notion, a retrospective study of equine actinobacillosis indicated that *A. equuli* subsp. *equuli* was isolated more frequently than hemolytic *Actinobacillus* spp. (presumably *A. equuli* subsp. *haemolyticus*) from the respiratory tract of foals and adult horses with respiratory disease manifestations, suggesting that *A. equuli* subsp. *equuli* can likely be primary pathogens under favorable conditions [12].

To enhance our knowledge about the potential invasion determinants and the virulence factors associated with *A. equuli* subsp. *equuli*, comprehensive genome analysis of the 4524 isolate was performed and compared with the reference *A. equuli* subsp. *equuli* ATCC 19392 strain available in GenBank. Infection of invasive species of Actinobacillus is a multifactorial process governed by various virulence factors, such as RTX toxins, LPS, CPS, OMPs, and biofilm, among others, to establish a productive infection in the host [2,20,21]. The RTX toxins are prevalent in *Actinobacillus* spp. and play an important role in their pathogenicity [34]. *Actinobacillus equuli* subsp. *haemolyticus* and *A. equuli* subsp. *equuli* differ by the presence or absence of an RTX, referred to as the *A. equuli* toxin (Aqx) encoded by the *aqx* gene, responsible for the hemolytic phenotype [34]. RTX toxins also contribute to the host specificity of the two bacterial species, *A. suis* and *A. equuli* subsp. *haemolyticus* containing the *apx* and the *aqx* genes, respectively. The hemolytic species *A. suis* had the *apxI* and *apxII* genes but not *aqx*, which was only found in strains of equine origin, suggesting that different hemolysins are associated with the hemolytic phenotype in these species [35,36]. In addition, the ApxI and ApxII were found to be more cytotoxic to swine lymphocytes. In contrast, the Aqx was more toxic to equine lymphocytes, indicating the host cell-specific activity of RTX [35]. Interestingly, *apxIB* gene encoding toxin exporter, but not *aqx*, was detected in the genome of the 4524 isolate, suggesting that this isolate could encode the protein for toxin secretion. Furthermore, the sequence of the toxin protein was highly similar to the *aqxB* gene of its hemolytic counterpart, *A. equuli* subsp. *haemolyticus*, the causative agent of fatal pulmonary hemorrhage in the adult horse [15]. Accordingly, we can speculate that the RTX toxin gene detected in *A. equuli* subsp. *equuli* isolate 4524 might have evolved to increase the host-specific cytotoxicity or transferred from its counterpart through the horizontal gene transfer mechanisms, resulting in increased virulence.

Endotoxin, associated with all gram-negative bacteria, damages endothelium, causing vascultilis and thrombosis [37]. Interestingly, several strain-specific putative surface polysaccharides that contribute to bacterial adherence, such as LPS (*glycosyltransferase family 2 protein* and *rfaF*) and CPS (*glycosyltransferase YkoT*) were also detected in the genome of *A. equuli* subsp. *equuli* 4524 isolate. Notably, the nucleotide and amino acid sequences of *wbaP* and *glf* genes of the 4524 isolate showed higher similarity to those of *A. suis* NCTC12996 and ATCC 33415 strains and *A. pleuropneumoniae* 3906 strain, respectively, compared to that of the reference *A. equuli* subsp. *equuli* ATCC 19392 strain. This observation suggested that the 4524 isolate may express the adherence molecules required for the firm adherence of the bacteria to the host cells, similar to those of invasive species, such as *A. suis* and *A. pleuropneumoniae* [38,39], resulting in enhancement of the pathogenicity of the 4524 strain; however, further investigation is needed.

Lastly, multiplication of the pathogenic microorganism within host tissues is essential for establishing infection. It partially depends on the ability of the pathogen to scavenge essential nutrients. Iron is essential for bacterial growth and regulating the expression of many virulence factors. The genome of the *A. equuli* subsp. *equuli* 4524 strain contained different high-affinity iron-acquisition systems to uptake iron from siderophores, transferrin, and hemoglobin proteins associated with TonB-dependent energy transduction. The *tonB* genes encode the proteins needed for the active transport of iron through the outer membrane [40]. The genome content of 4524 strain indicated the potential of the bacteria in transporting iron from transferrin, hemoglobin (*hgbA*), or ferric siderophore (*fhuBCD*) across the outer membrane depending on the exbB–exbD–TonB system [40,41]. The identified operon *fhuBCD* in the 4524 isolate might be involved in the translocation of ferric hydroxamate from the outer to the inner membrane (*fhuD*) and in the internalization of ferric hydroxamate (*fhuBC*) [41,42]. Taken together, *A. equuli subsp. equuli* 4524 isolate is genetically well-equipped to overcome iron shortages during infection. The results of genome mining suggested that infection by *A. equuli* subsp. *equuli* 4524 strain is a multifactorial process, and the putative virulence-associated genes may play a role in pathogenesis and disease phenotype in an adult horse.

## 5. Conclusions

Here, we report a case of an adult equine actinobacillosis associated with fibrinosuppurative and hemorrhagic pleuropneumonia caused by *A. equuli*. The phylogenetic analyses identified the isolate as *A. equuli* subsp*. equuli* owing to clustering together with *A. equuli* subsp. *equuli* ATCC 19392 reference strain. Further genome comparative analysis revealed the presence of seven unique virulence-associated genes in the *A. equuli* subsp. *equuli* 4524 isolate, but not in the reference strain, related to the pathogenesis and invasiveness in certain invasive species of *Actinobacillus* genus. Thus, we speculate that these unique features together with the putative virulence-associated genes (*glycosyltransferase family 2 protein*, *rfaF*, *glycosyltransferase YkoT*, *glf*, *wbaP*, and *apxIB*) related to the pathogenesis and invasiveness in certain invasive species of Actinobacillus genus, may be closely associated with the virulence phenotype observed in our clinical case. However further studies will be needed to compare 4524 and non-laboratory-adapted *A. equuli* subsp. *equuli* strains, other than the type strain, as well as *A. equuli* subsp. *haemolyticus* strains, to reveal more information on the functions of virulence-associated genes related to the pathogenesis of *A. equuli* subsp. *equuli.*

## Figures and Tables

**Figure 1 pathogens-12-00224-f001:**
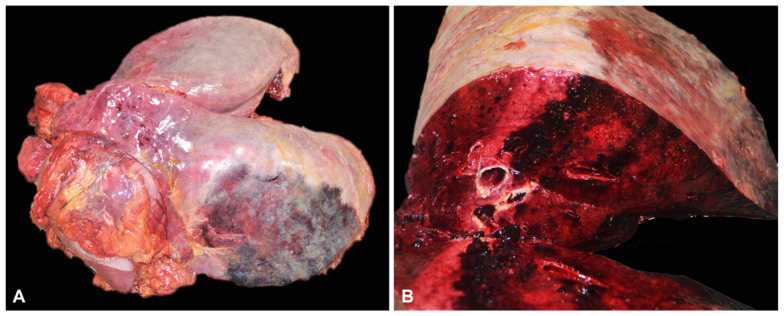
Gross findings in the lungs of the *Actinobacillus equuli*-infected horse. (**A**) The left lung is unilaterally affected by an extensive, well-demarcated dark red to black area of hemorrhage and necrosis coated by fibrinous exudate (hemorrhagic and fibrinous pleuropneumonia). (**B**) Left lung, cut surface. The cut surface shows an extensive and relatively well-delimited area of necrosis and hemorrhage.

**Figure 2 pathogens-12-00224-f002:**
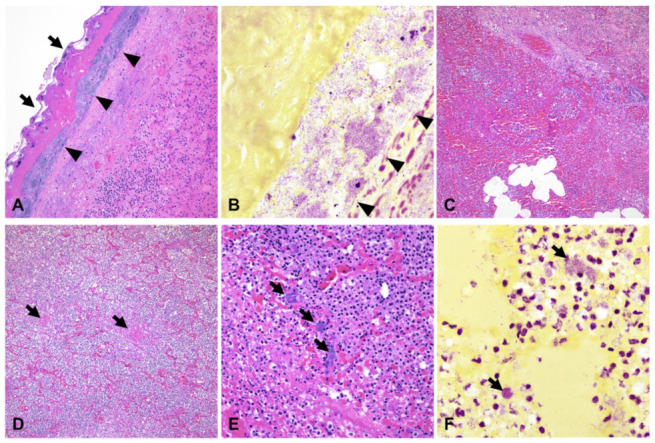
Microscopic features of *Actinobacillus equuli*-associated pleuropneumonia. (**A**) The pleura is extensively coated by a thick layer of polymerized fibrin (arrows) and myriad coccobacillary bacteria underlying the fibrinous exudate and extending into the subpleural interstitium (arrowheads). The underlying pulmonary parenchyma is affected by hemorrhage with loss of alveolar septa and replacement by numerous degenerate leukocytes. (**B**) The bacteria underlying the fibrinous exudate are gram-negative (arrowheads). (**C**,**D**) The pulmonary parenchyma is affected by extensive hemorrhage, fibrinous exudation, and necrosis and is infiltrated by numerous degenerate leukocytes. Within areas of less pronounced necrosis, alveolar spaces are filled with degenerate leukocytes (**D**), and the pulmonary vasculature is multifocally occluded by fibrin thrombi ((**D**), arrows). (**E**,**F**) Within affected areas of the parenchyma, there are multifocal extracellular and intracellular colonies of gram-negative (**F**) coccobacilli (arrows). (**A**,**C**,**D**,**E**); H&E, 200×, 100×, 100×, and 400×, respectively. (**B**,**F**); Gram stain, 1000×.

**Figure 3 pathogens-12-00224-f003:**
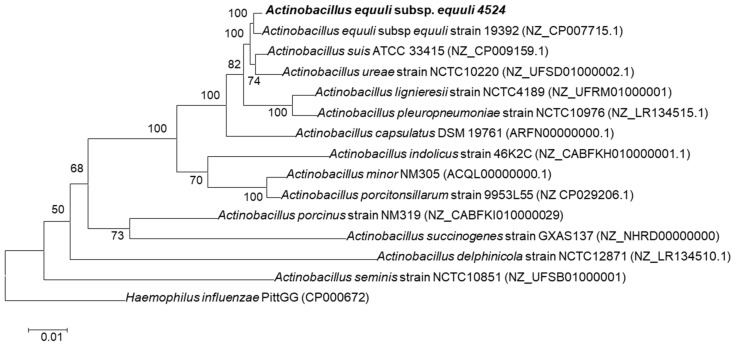
Maximum likelihood phylogenetic tree of the complete genome sequences of isolate 4524 and other *Actinobacillus* spp. based on REALPY. Numbers at nodes represent the percentages of occurrence of nodes in 1000 bootstrap trials. The *Haemophilus influenzae* PittGG (CP000672) was used as an outgroup.

**Figure 4 pathogens-12-00224-f004:**
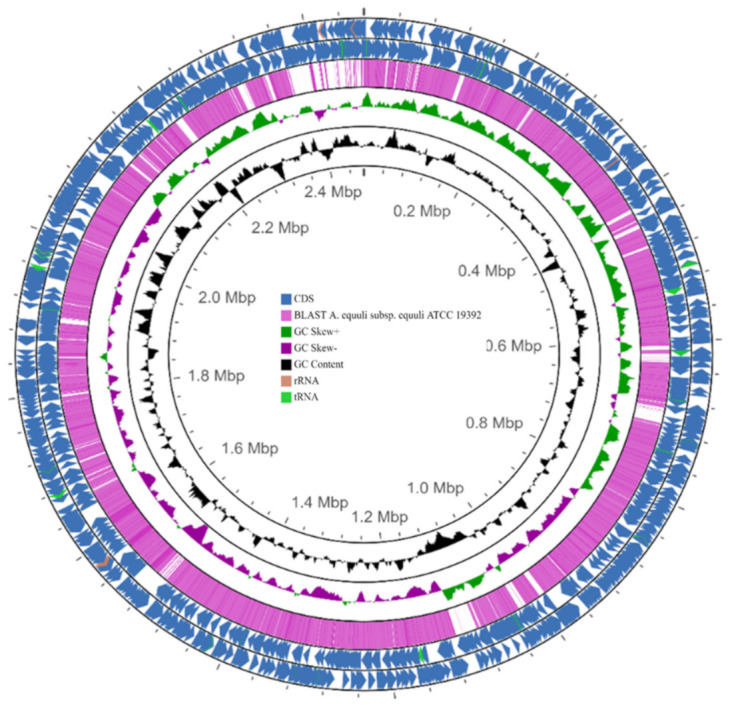
Circular map of the genome of *A. equuli* subsp. *equuli* 4524 strain generated using the CGView Server. From the outside to the center: coding sequences (CDSs) in the positive strand, CDSs in the reverse strand, BLASTn vs. *A. equuli* subsp. *equuli* ATCC 19392 (NZ_CP007715.1), GC skew, and GC content.

**Figure 5 pathogens-12-00224-f005:**
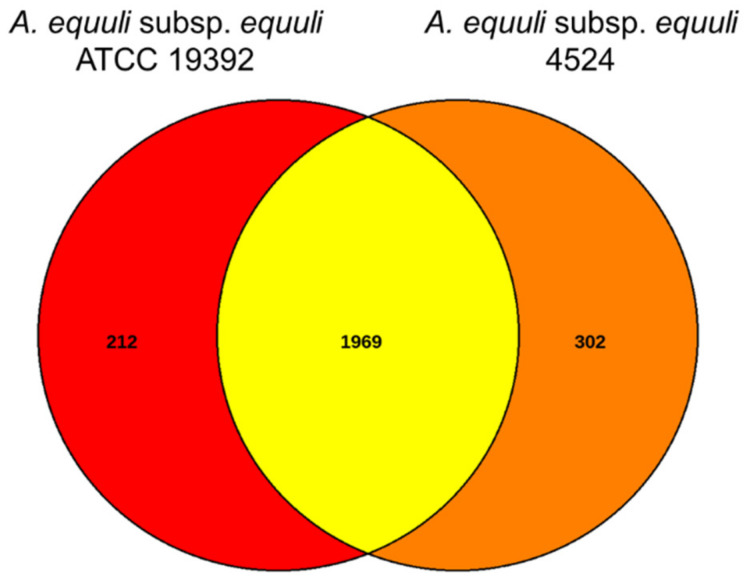
Venn diagram of the complete genomes of *A. equuli* subsp. *equuili* 4524 and ATCC 19392 strains.

**Table 1 pathogens-12-00224-t001:** Broth microdilution MIC results.

Antimicrobic	Result	Interpretation
Amikacin	=16	Susceptible
Ampicillin	≤0.25	No Interpretation
Azithromycin	≤0.25	No Interpretation
Cefazolin	≤4	No Interpretation
Ceftazidime	≤1	Susceptible
Ceftiofur	≤0.25	No Interpretation
Chloramphenicol	≤4	Susceptible
Clarithromycin	≤1	No Interpretation
Doxycycline	≤2	Susceptible
Enrofloxacin	≤0.25	No Interpretation
Erythromycin	=0.5	No Interpretation
Gentamicin	=8	Intermediate
Imipenem	≤1	Susceptible
Oxacillin	≤0.25	No Interpretation
Penicillin	=0.25	No Interpretation
Rifampicin	=4	No Interpretation
Tetracycline	≤2	No Interpretation
Ticarcillin	≤8	No Interpretation
Ticarcillin-Clavulanate	≤8	Susceptible
Trimethoprim-sulfamethoxazole	≤0.5	Susceptible

**Table 2 pathogens-12-00224-t002:** Genome statistics of *A. equuli* subsp. *equuli* 4524 isolate.

Attribute	Value
Genome size (bp)	2,464,726
Total genes	2362
Protein coding genes	2277
RNA genes	85
tRNA genes	61
tmRNA genes	1
rRNA genes	8
ncRNA genes	15
Genes assigned to COGs	1419
CRISPR repeats	2

**Table 3 pathogens-12-00224-t003:** The number of genes associated with general Cluster of Orthologous Group (COG) functional categories.

Code	Value	Description
J	203	Translation, ribosomal structure, and biogenesis
A	1	RNA processing and modification
K	83	Transcription
L	89	Replication, recombination, and repair
B	0	Chromatin structure and dynamics
D	33	Cell cycle control, cell division, chromosome partitioning
Y	0	Nuclear structure
V	30	Defense mechanisms
T	36	Signal transduction mechanisms
M	148	Cell wall/membrane/envelope biogenesis
N	6	Cell motility
Z	0	Cytoskeleton
W	1	Extracellular structures
U	28	Intracellular trafficking, secretion, and vesicular transport
O	88	Posttranslational modification, protein turnover, chaperones
C	104	Energy production and conversion
G	127	Carbohydrate transport and metabolism
E	168	Amino acid transport and metabolism
F	68	Nucleotide transport and metabolism
H	100	Coenzyme transport and metabolism
I	42	Lipid transport and metabolism
P	108	Inorganic ion transport and metabolism
Q	8	Secondary metabolites biosynthesis, transport, and catabolism
R	69	General function prediction only
X	4	Mobilome: prophages, transposons
S	78	Function unknown

**Table 4 pathogens-12-00224-t004:** Average nucleotide identity (ANI) and Genome-to-Genome Distance Calculation (GGDC), DNA-DNA hybridization (DDH) values between *A. equuli* 4524 and selected *Actinobacillus* spp.

Species	Strain	ANI (%)	GGDC	DDH	Prob. DDH ≥ 70%
* A. capsulatus *	DSM 19761	92.7	0.0739	49	16.6
* A. delphinicola *	NCTC12871	70.5	0.1742	25	0.01
*A. equuli* subsp. *equuli*	ATCC 19392	95.9	0.0418	66.1	70
* A. indolicus *	46K2C	74.5	0.2126	20.7	0
* A. lignieresii *	NCTC4189	86.8	0.1345	31.5	0.19
* A. minor *	NM305	76.4	0.1936	22.6	0
* A. pleuropneumoniae *	NCTC10976	86.5	0.1335	31.7	0.2
* A. porcinus *	NM319	73	0.1785	24.4	0.01
* A. porcitonsillarum *	9953L55	76.3	0.193	22.7	0
* A. seminis *	NCTC10851	71.8	0.1857	23.5	0
* A. succinogenes *	GXAS137	71.8	0.1847	23.7	0
* A. suis *	ATCC33415	93.2	0.0717	50	19
* A. ureae *	NCTC10220	92.7	0.0728	49.5	17.8

**Table 5 pathogens-12-00224-t005:** Shared and strain-specific virulence genes in *A. equuli* subsp. *equuli* 4524 and ATCC 19392 genomes.

Virulence Factor	Number of Shared Genes in 4524 and ATCC 19392	Number of Genes Specific to ATCC 19392	Number of Genes Specific to 4524
Iron acquisition	40	1	3
LPS	41	3	2
CPS	18	1	1
Miscellaneous	15	2	1
OMP	4	0	0
Salicylic acid	2	0	0
RTX toxin	1	0	0
Autotransporter	1	0	0
Total	122	7	7

## Data Availability

The complete genome sequence of *A. equuli* subsp. *equuli* 4524 isolate was deposited in NCBI GenBank under the accession number JAPHVQ000000000.

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
