# Peer review of "Pathological Features and Genomic Characterization of an Actinobacillus equuli subsp. equuli Bearing Unique Virulence-Associated Genes from an Adult Horse with Pleuropneumonia"

_pathogens, 2023, doi:10.3390/pathogens12020224_

Round 1

Reviewer 1 Report

See reviewer's opinion in the attached file

Reviewer 2 Report

Present article "Whole genome sequence and pathological features of Actinobacillus equuli subsp. equuli isolated from an adult horse with  pleuropneumonia" is fully in the scope of Pathogens, as it is not only a case report but also provides new pathogenicity-related genomic data on a Pasteurellaceae species, relatively unknown in this respect. Therefore, the title should reflect more appropriately the contents of the article and draw the attention of the reader to the new knowledge on the pathogen, having gained through the WGS. I suggest reconsidering the title.

Line 51: A. equuli subsp. equuli are largely unknown [2, 19, 20]. In the introduction, the authors should sentence more on the comparative description of the subsp. equuli and the subsp. haemolyticus, focusing on the differences in the phenotypic appearance (including classical bacteriological features), virulence factors, especially the presence/absence of RTX toxins.

Line 76 Bacteriology, biochemical testing, MALDI-TOF mass spectrometry, and antibiotic sensitivity.

Please improve the description of the classical bacteriology-based identification method.

The brief description of the MALDI-TOF method is missing.

Contrary to the title, the antibiotic sensitivity testing is absent in this section. Either delete or describe.

Section 3.3, line 187

Since A. suis did not fit with the case... - I suggest to completely rephrase the whole section. This is the key part of the article that gives a full proof to the identity of the pathogen responsible for the pathological lesions in this particular disease case. If he identification part is a house of cards, the whole article's credibility can be questionable. Thus, the identification has to include, in the logical order, colony morphology, appearance, haemolysis including the CAMP-test mentioned later and the classical biochemical tests performed. As it is described in the article of Christensen and Bisgaard (Vet. Mic. 99 (2004) 13–30), there are many similarities between A. suis and A. equuli subsp. equuli. A. equuli subsp. equuli can be differentiated from A. suis by the fermentation of D-Mannitol. A. equuli subsp. haemolyticus differs from A. equuli subsp. equuli in the presence of the haemolysis, which is caused by the Aqx toxin. These should be addressed in this part. The additional biochemical tests are to be performed if needed. The two subspecies can be discriminated by the presence of the Aqx toxin gene, which can, of course, be revealed with the WGS.

The distinction is important, since the haemolytic subspecies was reported to cause haemorrhagic pulmonary lesions in adult horses.

Regarding the MALDI-TOF identification, Kuhnert et al. (J.  Mic.  Meth.  89  (2012)  1–7) mentions that there is a high similarity between A. equuli and A. suis, thus A. equuli can be misidentified as A. suis. This was confirmed by Uchida Fujii et al (J. Vet. Med. Sci. 81(9) (2019), stating that the MALDI-TOF method can misidentify subsp. haemolyticus as A. suis. 

These underline the importance of the solid prior bacteriological identification.

Section 3.3 There is no antibiotic resistance data in the results section either, contrary to what the title states.

Unfortunately Table 3 can not include the data of the the type strain of A. equuli subsp. haemolyticus, NCTC 13195, which would be interesting in the genome distance comparison. Unfortunately, there is no full sequence available of this strain yet.

Line 250, regarding the RTX toxin genes

Subsp. equuli and haemolyticus and subsp. equuli differ in the presence of the Aqx toxin gene, whereas A. suis has the Apx gene, which is not in the former two bacteria, as it is mentioned in the discussion part. How is this related to your WGS data?

Line 264&269: apxIb encodes the toxin exporter. It is to be  described in the discussion too, in Line 311.

Lines 315-318: ??? The toxin exporter gene is certainly there, but how is it related to the one of A. subsp. haemolyticus or A. suis? Is there a proof of horizontal gene transfer?

Section 5. It is to be mentioned that further studies needed to compare 4524 and  non-laboratory-adapted A. equuli subsp. equuli strains, other than the type strain, as well as A. equuli subsp. haemolyticus strains, to reveal more information on the functions of virulence-associated genes, needed for the pathogenesis of A. equuli subsp. equuli.

Reviewer 3 Report

Thank you for the opportunity to review this detailed and well described case study of a clinical case of Actinobacillus equuli subsp. equuli infection causing pleuropneumonia in an adult horse. As you explained, disease due to A. equuli subsp. equuli is uncommon in adult horses. A full-length genome sequence of the isolate was obtained and characterized. The purposes of the characterization and analysis of the genome were to improve the understanding of potential invasion determinants and virulence factors associated with pathogenicity and disease phenotype of A. equuli subsp. equuli.

 The methods for the necropsy, sample collection, histopathological processing and microbiological investigation were in general clear and concise.

The methods for DNA extraction, whole genome sequencing and assembly, genome annotation and analysis and phylogenetic analysis of Actinobacillus equuli subsp. equuli were well described and valid.

The references used in the paper were relevant.

There are very few recently published case studies (within the past 5-7 years) reporting Actinobacillus equuli subsp equuli in older horses and assessments of putative virulence genes and invasive determinant genes following whole genome sequencing of Actinobacillus equuli subsp equuli. In this manuscript, genome comparative analysis between  Actinobacillus equuli subsp. equuli strain 4524 and  Actinobacillus equuli subsp. equuli ATCC 19392  was completed. Seven putative virulence factor genes of Actinobacillus genus (including glycosyltransferase family 2 protein, rfaF, glycosyltransferase YkoT, glf, 356 wbaP and apxIB) were found only in  Actinobacillus equuli subsp. equuli strain 4524. These findings supported the assessment that these putative virulence factors may be closely associated with the virulence phenotype related to the pleuropneumonia in the adult American paint horse. The need for mechanistic studies to demonstrate the roles of these virulence factors in the pathogenicity of A. equuli subsp. equuli was clearly noted in the discussion.

The manuscript was clearly written, technically sound and scientifically valid. This manuscript will add to the understanding of potential virulence factors for A. equuli subsp. equuli and provide the basis for future research to clarify the roles of those virulence factors. In my opinion, the manuscript is suitable for publication after some minor suggested changes (listed below).

Suggested corrections

Line 76 3.3. Bacteriology, MALDI-TOF mass spectrometry, and antibiotic sensitivity.

There are no methods for antibiotic sensitivity testing in the paragraph below this heading. Can the methods please be entered.

Line 187: 3.3. Bacteriology, MALDI-TOF mass spectrometry, and antibiotic sensitivity.

There are no results for antibiotic sensitivity testing. Can they please be entered.

Based on biochemical and phenotypic results reported in the paragraph, there is insufficient information to support the identification of Actinobacillus equuli.

Can the results of additional testing e.g.,

NAD-independent growth,

fermentation of sucrose, mannitol, galactose, lactose, maltose, mannose, melibiose, trehalose, raffinose, glycerol,

reduction of nitrate and  production of α-galactosidase, α-glucosidase, β-xylosidase, urease, and oxidase be included in a supplementary table ?

The relevant tests could also be included in the method section.

Figure 1A line 158

There is an arrow noted in the descriptions for Figure 1A but there is no arrow in the photo. The photo does not need an arrow. Can the reference to the arrow be deleted ?

Figures 2B and 2F

Gram negative bacteria are described in both photomicrographs but the lack of magnification limits objective assessment by the reader. Can these photomicrographs be replaced with photos at 1000x (oil objective) to clearly show the Gram negative bacteria ?

Lines 320-323

Please review the grammar for this sentence:

“Interestingly, several strain-specific putative surface ….”

References

Based on the MDPI guide for authors (https://www.mdpi.com/authors/references)  references should include all the names of the authors. Can the references be corrected to include all the author names rather than “et al”?
